# Imidacloprid and Its Bioactive Metabolite, Desnitro-Imidacloprid, Differentially Affect Ovarian Antral Follicle Growth, Morphology, and Hormone Synthesis In Vitro

**DOI:** 10.3390/toxics11040349

**Published:** 2023-04-07

**Authors:** Vasiliki E. Mourikes, Ramsés Santacruz Márquez, Ashley Deviney, Alison M. Neff, Mary J. Laws, Jodi A. Flaws

**Affiliations:** Department of Comparative Biosciences, University of Illinois Urbana-Champaign, Urbana, IL 61802, USA

**Keywords:** ovary, ovarian follicles, antral follicles, pesticides, neonicotinoids, imidacloprid, desnitro-imidacloprid, reproductive toxicology, pesticide toxicology

## Abstract

Imidacloprid is a neonicotinoid pesticide used in large-scale agricultural systems, home gardens, and veterinary pharmaceuticals. Imidacloprid is a small molecule that is more water-soluble than other insecticides, increasing the likelihood of large-scale environmental accumulation and chronic exposure of non-targeted species. Imidacloprid can be converted to the bioactive metabolite desnitro-imidacloprid in the environment and body. Little is known about the mechanisms by which imidacloprid and desnitro-imidacloprid induce ovarian toxicity. Thus, we tested the hypothesis that imidacloprid and desnitro-imidacloprid differentially affect antral follicle growth and steroidogenesis in vitro. Antral follicles were dissected from the ovaries of CD-1 mice and cultured in media containing vehicle control or 0.2 µg/mL–200 µg/mL of imidacloprid or desnitro-imidacloprid for 96 h. Follicle morphology was monitored, and follicle size was measured every 24 h. At the end of the culture periods, media were used to quantify follicular hormone levels, and follicles were used for gene expression analysis of steroidogenic regulators, hormone receptors, and apoptotic factors. Imidacloprid did not affect follicle growth or morphology compared to the control. Desnitro-imidacloprid inhibited follicle growth and caused follicles to rupture in culture compared to the control. Imidacloprid increased progesterone, whereas desnitro-imidacloprid decreased testosterone and progesterone compared to the control. Desnitro-imidacloprid also changed estradiol compared to the control. At 48 h, IMI decreased the expression of *Star*, *Cyp17a1*, *Hsd17b1*, *Cyp19a1*, and *Esr2* and increased the expression of *Cyp11a1*, *Cyp19a1*, *Bax*, and *Bcl2* compared to the control. IMI also changed the expression of *Esr1* compared to the control. At 48 h, DNI decreased the expression of *Cyp11a1*, *Cyp17a1*, *Hsd3b1*, *Cyp19a1*, and *Esr1* and increased the expression of *Cyp11a1*, *Hsd3b1*, and *Bax* compared to the control. At 72 h of culture, IMI significantly decreased the expression of *Cyp19a1* and increased the expression of *Star* and *Hsd17b1* compared to the control. At 72 h, DNI significantly decreased the expression of *Cyp11a1*, *Cyp17a1*, *Hsd3b1*, and *Bax* and increased the expression of *Esr1* and *Esr2*. At 96 h, IMI decreased the expression of *Hsd3b1*, *Cyp19a1*, *Esr1*, *Bax*, and *Bcl2* compared to the control. At 96 h, DNI decreased the expression of *Cyp17a1*, *Bax*, and *Bcl2* and increased the expression of *Cyp11a1*, *Hsd3b1*, and *Bax* compared to the control. Together, these data suggest mouse antral follicles are targets of neonicotinoid toxicity, and the mechanisms of toxicity differ between parent compounds and metabolites.

## 1. Introduction

Neonicotinoid pesticides have become some of the most popular agricultural chemicals in the world as they are actively being chosen as replacements for organophosphorus, organochlorine, and carbamate pesticides [1]. They are synthetic nicotine derivatives that act as systemic neurotoxicants by binding nicotinic acetylcholine receptors (nAChR) in the nervous system [2]. Neonicotinoids have a broad target base that includes both sucking and chewing insects, making them useful in many industries [3]. Imidacloprid (IMI) is the first synthesized neonicotinoid used in large-scale agricultural systems, sold commercially for use in private homes, and found in veterinary pharmaceuticals (e.g., flea and tick preventatives). In part, neonicotinoids are popular because they can be used as crop seed treatments. Although seed treatments are cost-efficient for pest management, they come with other negative consequences [4]. Insecticides used as seed treatments cannot be washed or peeled off produce because they spread systemically throughout the crop as it matures [4]. Additionally, the chemical leaches out of the seed, contaminating agricultural lands and water systems [5]. This is especially concerning because neonicotinoids are small molecules that are more water-soluble than other insecticides, increasing their likelihood of environmental accumulation [6]. In fact, IMI has been detected in drinking water sources and finished drinking water around the world [7,8,9,10,11,12]. Imidacloprid’s rising popularity as an insecticide, coupled with its ubiquitous presence in the environment, results in the chronic exposure of non-target species. Humans are primarily exposed to IMI through the ingestion of contaminated food and water [1].

To date, exposure to IMI has not been considered a public health concern. In fact, neonicotinoids are widely considered safe for mammals, hence their use as flea and tick preventatives in companion animals. Due to their high affinity for insect nAChRs and comparatively low affinity for mammalian nAChRs, neonicotinoids are regarded as insect-selective and thus safer than other pesticides. Although IMI is a weak nAChR agonist in mammals, it can be converted in the body to more toxic metabolites, namely desnitro-imidacloprid (DNI) [13,14]. In mammals, IMI is readily absorbed through the gastrointestinal tract. Once it reaches the liver, IMI is converted to a variety of intermediate metabolites by cytochrome p450 monooxygenases (CYP) and aldehyde oxidases (AOX) [15]. Importantly, the nitroreduction of IMI by AOX1 yields the bioactive metabolite DNI. DNI is estimated to be significantly more toxic to mammals than IMI because of its higher affinity for mammalian nAChRs than IMI [16,17,18,19]. DNI is further broken down to the terminal metabolites 6-chloronicotinic acid (6CNA) and 6-hydroxynicotinic acid (6HNA), which are excreted both in urine and feces. Tissue disposition experiments conducted in Wister rats indicate that IMI, 6CNA, and 6HNA remained in the body for at least 48 h after a single oral exposure to IMI (20 mg/kg). Furthermore, all three compounds were detectable in the ovaries of the female animals within hours of exposure and reached peak concentrations (IMI 0.20 µg/g, 6CNA 0.40 µg/g, 6HNA 0.65 µg/g) at 12 h post-exposure [20]. 

The structural diversity and unique biochemical processes that take place in the ovary provide many possible targets for environmental toxicants, especially pesticides. The ovary is crucial for reproductive and systemic health. The functional structures within the ovary are follicles at various stages of maturity and the corpora lutea, held together by stromal tissue and surrounded by a layer of germinal epithelium [21]. The most mature follicles, called antral follicles, are critically important because only antral follicles release eggs for fertilization, and they are the primary producers of sex steroid hormones (estradiol, progesterone, and testosterone), which maintain reproductive, cardiovascular, brain, and skeletal health [22]. Antral follicles synthesize the sex steroid hormones through the concerted efforts of steroidogenic regulators in theca and granulosa cells. Cholesterol is brought into theca cells via steroidogenic acute regulatory protein (STAR). Cholesterol is converted to pregnenolone via cytochrome P450 cholesterol side-chain cleavage (CYP11A1). Pregnenolone can be converted to progesterone by 3β-hydroxysteroid dehydrogenase 1 (HSD3B1) or to dehydroepiandrosterone by cytochrome P450 steroid 17-a-hydroxylase 1 (CYP17A1). Progesterone and dehydroepiandrosterone are converted to androstenedione by HSD3B1 or CYP17A1. Androstenedione is converted to testosterone by 17β-hydroxysteroid dehydrogenase 1 (HSD17B1). The androgens synthesized in theca cells are secreted into granulosa cells, where they are converted to estradiol by CYP19A1. Progesterone is synthesized in theca cells as an intermediate in estrogen biosynthesis, but it can also be synthesized in granulosa cells via CYP11A1 and HSD3B1. 

Antral follicles rely on estrogen signaling to continue growing [23]. Thus, hormone synthesis and antral follicle growth are physiologically interconnected. Antral follicles express both estrogen receptor 1 and estrogen receptor 2 (ESR1 and ESR2), which allow the follicles to respond to hormone signals and mediate growth [23]. In addition to hormone signaling, antral follicle growth is heavily influenced by each follicle’s ability to maintain the proper balance of pro-apoptotic and anti-apoptotic signals. Specifically, the pro-apoptotic factor (i.e., Bcl2-associated X protein (BAX)) and the anti-apoptotic factor (i.e., B cell lymphoma 2 (BCL2)) are critical regulators of follicle growth [24,25]. The expression and activity of steroidogenic regulators, estrogen receptors, and apoptotic factors are key determinants of antral follicle health. In fact, many ovarian toxicants have been shown to target the transcriptional and translational regulation of these genes [26,27,28,29,30]. 

Some adverse reproductive outcomes of IMI have been demonstrated in laboratory rodents. Adult female rats exposed to IMI (20 mg/kg/day) for 90 days had decreased ovarian weights, significant morphological changes in preantral and antral follicles, increased numbers of atretic follicles, increased follicle stimulating hormone levels, and decreased luteinizing hormone and progesterone levels compared to the control rats [31,32]. Further, IMI decreased ovarian expression of the antioxidants glutathione, superoxide dismutase, catalase, and glutathione peroxidase [31,32]. These data suggest that IMI is a reproductive toxicant in rats; however, little information is available on the effects of IMI on antral follicle growth and steroidogenesis. 

Little is known about the role that DNI (a bioactive metabolite) plays in IMI-induced ovarian toxicity. Among many IMI metabolites, we prioritized the use of DNI in our experiments because of its potential to cause mammalian toxicity. In the body, IMI is hydroxylated to DNI, an intermediate metabolite that is estimated to be more toxic to mammals than IMI [16,18,19]. Therefore, the objective of this study was to test the hypothesis that IMI and DNI differentially affect mouse antral follicle growth and steroidogenesis in vitro. To test this hypothesis, we compared the effects of direct IMI and DNI exposure on follicle growth, morphology, sex steroid hormone production, and the expression of genes that regulate steroidogenesis and follicle growth in vitro. 

## 2. Materials and Methods

### 2.1. Chemicals

The chemicals used in this study were dimethyl sulfoxide (DMSO), IMI, and DNI, which were all purchased from Sigma-Aldrich (St. Louis, MO, USA). IMI and DNI are PESTANAL analytical standards with purities of ≥98.0% and ≥99.0%, respectively. Stock solutions of IMI were prepared in DMSO, and stock solutions of DNI were prepared in ultrapure cell culture water based on each chemical’s solubility. Treatment solutions of IMI and DNI were prepared by diluting the stocks in their respective solvents such that equal volumes of each treatment (0.75 µL) could be added per milliliter of culture media. Treatment solutions (0.267, 2.67, 26.7, and 267 µg/mL) were aliquoted and stored at −80 °C. On the first day of each culture, aliquots were thawed in a 37 °C bead bath and added to supplemented culture media for final working concentrations of 0.2, 2, 20, and 200 µg/mL. We found that changes in the ambient temperature affected the solubility of the highest concentration of IMI, so all treated media were sonicated in a solid state/ultrasonic water bath (model FS-14, Fisher Scientific, Waltham, MA, USA) for 30 min or until the IMI was completely dissolved in the media. Neonicotinoids are used in diverse industries (agriculture, home gardening, and veterinary medicine). As such, human exposure varies drastically based on geographical location, season, and occupation. Some of the selected doses were based on human biomonitoring literature from Asia, Europe, and North America and the Centers for Disease Control and Prevention’s National Health and Nutrition Examination Survey (NHANES), which is one of the largest and most inclusive exposure databases in the United States [1,33,34,35,36,37]. It is not possible to extrapolate exactly how much IMI or DNI reaches a single ovarian follicle. Thus, we also opted to use a wide range of doses that surpass the spectrum of human exposure and allow us to identify dose response relationships. The lowest tested concentrations reflect those measured in rats treated with 20 mg/kg IMI [20]. Although the highest dose (200 µg/mL) exceeds ovarian exposure, we included this dose to elicit an exaggerated response to IMI and DNI so we could differentiate their effects on antral follicle physiology. 

### 2.2. Animals

CD-1 mice of reproductive age (postnatal days (PND) 32–42) were purchased from Charles River Laboratories (Wilmington, MA, USA). The mice were housed in the animal facility located at the University of Illinois College of Veterinary Medicine, where they were given a 1–2 week acclimation period prior to use. The mice were kept under 12 h light-dark cycles at 22 ± 1 °C and were provided food and water ad libitum. All procedures involving animal care, euthanasia, and tissue collection were approved by the Institutional Animal Use and Care Committee at the University of Illinois at Urbana-Champaign. We elected to use the CD-1 mouse as our model because mice have been used in thousands of studies on the effects of chemicals on female reproduction, and we wanted to be able to compare our results on the effects of IMI and DNI on the mouse follicle with the results from other studies examining different chemicals [26,27,28,29,30]. Further, mouse ovarian follicles resemble ovarian follicles from other species because many of the genes and proteins expressed in ovarian follicles are conserved across mammalian species.

### 2.3. Antral Follicle Culture

Antral follicles (220 to 400 µm) were manually isolated from the ovaries of CD-1 mice (PND 32–42) using watchmaker’s forceps under a dissecting microscope. Each follicle was placed in a single well of a 96-well plate containing 150 µL of supplemented media containing either vehicle control, IMI (0.2, 2, 20, and 200 µg/mL), or DNI (0.2, 2, 20, and 200 µg/mL). DMSO was used as the vehicle control in the IMI cultures, and ultrapure water was used as the vehicle control in the DNI cultures. The supplemented media contained α-MEM (Gibco, Waltham, MA, USA) with 1% ITS (10 ng/mL insulin, 5.5 ng/mL transferrin, and 5 ng/mL selenium, Sigma-Aldrich), 100 U/mL penicillin (Sigma-Aldrich), 100 mg/mL streptomycin (Sigma-Aldrich), 5 IU/mL recombinant follicle-stimulating hormone (Dr. A.F. Parlow, National Hormone and Peptide Program, Harbor-UCLA Medical Center, Torrance, CA, USA), and 5% fetal bovine serum (Atlanta Biologicals, Lawrenceville, GA, USA). Follicles were cultured for 48, 72, or 96 h in a 37 °C incubator supplying 5% CO_2_. Each culture was repeated 3–5 times, with 8–12 follicles per treatment group per culture. At the end of the culture period, follicles were collected for gene expression analysis and media were collected for steroid hormone quantification. The follicles from each treatment group were collected in a 1.5 mL Eppendorf tube, snap frozen in liquid nitrogen, and stored at −80 °C. The media were removed from the plate and stored in 2 mL Eppendorf tubes at −80 °C.

### 2.4. Antral Follicle Growth and Rupture

Follicle growth was monitored over a 96-hour culture period. Follicles were measured immediately after they were plated and again every 24 h through the end of the culture period. The size of each follicle was measured using the diameter across perpendicular axes with an inverted microscope equipped with a calibrated ocular micrometer. Within each treatment group, the sizes of each follicle were averaged and converted to a percentage, where the initial follicle size (0 h) was defined as 100%. The measurements of follicles that ruptured during the culture period were not included in the growth analysis, and follicle rupture was quantified separately. The ruptured follicles and the media from these follicles were included in the tissue and media collection at the end of each culture period.

### 2.5. Steroid Hormone Quantification 

The steroid hormones estradiol, testosterone, and progesterone were quantified in the culture media collected at the end of each culture period (48 h, 72 h, and 96 h). The media from the 48 h and 72 h cultures were diluted 5-fold in steroid-free serum, and the media from the 96 h cultures were diluted 10-fold in the same diluent. The media were subjected to enzyme-linked immunosorbent assays (DRG International Inc., Springfield, NJ, USA) by following the manufacturer’s protocol. The analytical sensitivity of the assays for each hormone was 10.6 pg/mL for estradiol, 0.083 ng/mL for testosterone, and 0.034 ng/mL for progesterone. Absorbances were read on a BioTek Synergy LX multi-mode reader, while a BioTek Gen5 microplate reader and Imager software were used to generate standard curves and calculate hormone concentrations based on protocol-specific absorbances.

### 2.6. Gene Expression

Expression of relevant genes was quantified in the antral follicles that were snap-frozen at the end of each culture period (48 h, 72 h, and 96 h). Total RNA was isolated from the antral follicles using the RNeasy Micro Kit (Qiagen Inc., Valencia, CA, USA) according to the manufacturer’s protocol. RNA was eluted in the RNase-free water, and the concentration was determined using a NanoDrop (Nanodrop Technologies, Inc., Wilmington, DE, USA). Total RNA (300 ng) was reverse transcribed to complementary cDNA using the iScript RT kit (Bio-Rad Laboratories, Inc., Hercules, CA, USA) according to the manufacturer’s protocol. Quantitative polymerase chain reaction (qPCR) analysis was performed using cDNA from 1.67 ng total RNA, 0.75 pmol/µL of each forward and reverse primer, and 1x SsoFastEvaGreen dye (Bio-Rad Laboratories) in 10 µL reaction volumes. Gene expression was quantified using the CFX96 Real-Time Detection System (Bio-Rad Laboratories) and CFX Manager Software. The qPCR protocol began with incubation at 95 °C for 5 min. This was followed by 35 cycles at 95 °C for 10 s, 60 °C for 10 s, and 72 °C for 10 s. Melting from 65 °C to 95 °C was followed by extension at 72 °C for 2 min. Melting temperature graphs, standard curves, and threshold cycle (Ct) values were acquired for each gene analyzed. The expression data were normalized to the corresponding values for the housekeeping gene (*βactin*). Individual relative fold changes were calculated by the Pfaffl method, and then fold changes for each treatment were calculated and compared to the control for the corresponding culture. Lyophilized primers (sequences in Table 1) were purchased from Integrated DNA Technologies (Coralville, IA, USA) and dissolved in DNase- and RNase-free water. In this study, we analyzed the expression of the following genes: beta actin (*βactin*), steroidogenic acute regulatory protein (*Star*), cytochrome P450 cholesterol side-chain cleavage (*Cyp11a1*), cytochrome P450 steroid 17-a-hydroxylase 1 (*Cyp17a1*), cytochrome P450 aromatase (*Cyp19a1*), 3b-hydroxysteroid dehydrogenase 1 (*Hsd3b1*), 17b-hydroxysteroid dehydrogenase 1 (*Hsd17b1*), estrogen receptor 1 (*Esr1*), estrogen receptor 2 (*Esr2*), B cell lymphoma 2 (*Bcl2*), and bcl2-associated X protein (*Bax*).

### 2.7. Statistical Analyses

All data analyses were performed using SPSS statistical software (SPSS Inc., Chicago, Illinois). Data were expressed as means ± SEM (standard error of the mean) for three to five separate cultures per timepoint. Multiple comparisons between normally distributed experimental groups were made using one-way analysis of variance (ANOVA), followed by Dunnett’s post hoc comparison if equal variances were assumed or Games–Howell post hoc comparisons if equal variances were not assumed. If data were not normally distributed, comparison between two groups was performed using Mann–Whitney U 2-independent sample tests. Statistical significance was assigned at *p* ≤ 0.05. 

## 3. Results

### 3.1. Effects of IMI and DNI on Antral Follicle Growth and Follicle Rupture

IMI did not affect antral follicle growth (Figure 1A) or cause more follicles to rupture compared to the control. DNI at lower concentrations (0.2–20 µg/mL) did not significantly affect follicle growth compared to the control (Figure 1B), whereas the highest concentration of DNI (200 µg/mL) decreased antral follicle growth compared to the control, beginning at 24 h and continuing through 96 h of culture (Figure 1, *n* = 3–5, *p* < 0.05). DNI (0.2–200 µg/mL) also caused more antral follicles to rupture compared to the control, beginning at 24 h and continuing through 96 h of culture (Figure 2, *n* = 3–5, *p* < 0.05).

### 3.2. Effects of IMI and DNI on Sex Steroid Hormone Levels

After 48 h of culture, IMI did not affect sex steroid hormone levels compared to the control (Figure 3A). In contrast, DNI significantly increased estradiol levels (0.2 µg/mL) and decreased testosterone (200 µg/mL) and estradiol (200 µg/mL) levels compared to the control (Figure 3B, *n* = 3–5, *p* < 0.05). After 72 h of culture, IMI significantly increased progesterone levels (200 µg/mL) compared to the control, whereas DNI significantly decreased progesterone levels (200 µg/mL) compared to the control (Figure 4, *n* = 3–5, *p* < 0.05). After 96 h of culture, IMI significantly increased progesterone levels (200 µg/mL) compared to the control (Figure 5A, *n* = 3–5, *p* < 0.05). At this timepoint, DNI (200 µg/mL) significantly decreased testosterone, progesterone, and estradiol levels compared to the control (Figure 5B, *n* = 3–5, *p* < 0.05).

### 3.3. Effects of IMI and DNI on Gene Expression

#### 3.3.1. Steroidogenic Regulators 

After 48 h of culture, IMI significantly decreased the expression of *Star* (2 µg/mL), *Cyp17a1* (20 and 200 µg/mL), *Hsd17b1* (20 µg/mL), and *Cyp19a1* (0.2 µg/mL), and it significantly increased the expression of *Cyp11a1* (20 µg/mL) and *Cyp19a1* (2 µg/mL) compared to the control (Figure 6, *n* = 3–5, *p* < 0.05). At this timepoint, DNI significantly decreased the expression of *Cyp11a1* (200 µg/mL), *Cyp17a1* (2 and 20 µg/mL), *Hsd3b1* (2 µg/mL), and *Cyp19a1* (200 µg/mL), and it increased the expression of *Cyp11a1* (0.2 µg/mL) and *Hsd3b1* (20 µg/mL) compared to the control (Figure 6, *n* = 3–5, *p* < 0.05).

After 72 h of culture, IMI significantly decreased the expression of *Cyp19a1* (20 µg/mL) and increased the expression of *Star* (200 µg/mL) and *Hsd17b1* (200 µg/mL) compared to the control (Figure 7, *n* = 3–5, *p* < 0.05). At this timepoint, DNI significantly decreased the expression of *Cyp11a1* (0.2 µg/mL), *Cyp17a1* (20 µg/mL), and *Hsd3b1* (0.2 and 20 µg/mL), compared to the control (Figure 7, *n* = 3–5, *p* < 0.05). 

After 96 h in culture, IMI decreased the expression of *Hsd3b1* (0.2 µg/mL) and *Cyp19a1* (0.2 µg/mL) compared to the control (Figure 8, *n* = 3–5, *p* < 0.05). At this timepoint, DNI decreased the expression of *Cyp17a1* (200 µg/mL) and increased the expression of *Cyp11a1* (200 µg/mL) and *Hsd3b1* (200 µg/mL) compared to the control (Figure 8, *n* = 3–5, *p* < 0.05).

#### 3.3.2. Estrogen Receptors 

After 48 h of culture, IMI significantly increased the expression of *Esr1* (0.2 µg/mL) and decreased the expression of *Esr1* (200 µg/mL) and *Esr2* (0.2 µg/mL) compared to the control (Figure 9, *n* = 3–5, *p* < 0.05). In contrast, DNI significantly decreased the expression of *Esr1* (2 µg/mL) compared to the control (Figure 9A, *n* = 3–5, *p* < 0.05). After 72 h of culture, DNI significantly increased the expression of *Esr1* (0.2 and 20 µg/mL) and decreased the expression of *Esr2* (200 µg/mL) compared to the control (Figure 9B, *n* = 3–5, *p* < 0.05). After 96 h of culture, IMI significantly decreased the expression of *Esr1* (0.2 and 200 µg/mL) compared to the control (Figure 9C, *n* = 3–5, *p* < 0.05).

#### 3.3.3. Apoptotic Factors 

After 48 h of culture, IMI decreased the expression of *Bax* (20 µg/mL) and *Bcl2* (20 µg/mL), whereas DNI increased the expression of *Bax* (20 and 200 µg/mL) compared to the control (Figure 10A, *n* = 3–5, *p* < 0.05). After 72 h of culture, IMI did not affect the expression of *Bax*, but DNI decreased the expression of *Bax* (0.2 and 20 µg/mL) compared to the control (Figure 10B, *n* = 3–5, *p* < 0.05). After 96 h of culture, IMI decreased the expression of *Bax* (0.2 µg/mL) and *Bcl2* (0.2 and 200 µg/mL) (Figure 10C, *n* = 3–5, *p* < 0.05). In contrast, DNI decreased the expression of *Bax* (0.2 µg/mL) and *Bcl2* (20 µg/mL) and increased the expression of *Bax* (200 µg/mL) compared to the control (Figure 10C, *n* = 3–5, *p* < 0.05).

## 4. Discussion

In this study, we compared the effects of IMI and its bioactive metabolite DNI on mouse antral follicle growth, morphology, steroidogenesis, and the expression of genes that regulate steroidogenesis and follicle growth using an in vitro culture system. To our knowledge, we are the first to investigate the effects of any neonicotinoids or their metabolites on mouse antral follicles in vitro. Thus, we elected to use a culture system where we could directly expose antral follicles to IMI and DNI and compare their effects on follicular health. 

We found that IMI did not affect follicle growth, whereas DNI inhibited follicle growth as soon as 24 h in culture and continued to do so at 48, 72, and 96 h in culture. We also found that the follicles exposed to all concentrations of DNI ruptured in the culture. The reasons for the different effects of IMI and DNI on antral follicles likely stem from differences in their ability to bind nAChRs, leading to differences in toxicity [16]. nAChRs are membrane-bound ion channels that form when five subunits come together. The alpha subunit is the most important one because it is where ligands bind and because it controls when the channel opens and closes. The ligand-binding pocket of the alpha subunit is composed of a highly conserved core of uncharged aromatic amino acid residues. IMI, like other neonicotinoids, was synthesized with an electronegative sidechain that prevents it from forming stable interactions in the ligand binding pocket of the alpha subunit [16]. When IMI is reduced to DNI, the electronegative side chain is protonated, which allows it to bind the alpha subunit in a stable manner with high affinity [16,18,19].

Another indicator of follicle health in vitro is sex steroid hormone synthesis. Thus, we examined the effects of IMI and DNI on steroid hormone levels and the expression of steroidogenic regulators. Sex steroid hormone synthesis begins when cholesterol is brought into theca cells via the transport protein STAR. Within theca cells, cholesterol is converted to testosterone, which diffuses into neighboring granulosa cells, where it is converted to estradiol. Progesterone is an intermediate in testosterone biosynthesis, and it can also be synthesized in granulosa cells independently. In vivo, ovarian hormone secretion is regulated through the hypothalamic-pituitary-gonadal axis. Toxicant-induced changes to serum sex steroid hormone levels can originate from the hypothalamus, pituitary, or ovary. In our culture system, ovarian follicles were supplemented with FSH to stimulate follicle growth and hormone production, thus eliminating the influence of the hypothalamus or the pituitary on follicular hormone secretion.

We found that IMI increased progesterone levels at 72 h and 96 h compared to the control. This finding differed from an in vivo study, which reported that adult female rats exposed to IMI for 90 days had decreased serum progesterone levels compared to control rats [31]. It is important to note that serum hormone levels vary drastically throughout the estrous cycle, and estrous cyclicity was not controlled in the published in vivo study [31,32]. Therefore, it is not possible to determine if the reported changes in hormone levels were due to IMI exposure or estrous cyclicity. Although IMI disrupted the expression of steroidogenic regulators in our experiment, the changes do not explain the significantly increased progesterone levels in the culture media. This may indicate that IMI is interfering with the protein expression or enzymatic activity of these regulators instead of their transcription. Another possibility is that IMI is inhibiting the follicular metabolism of progesterone, resulting in a buildup in the culture media.

The highest concentration of DNI decreased progesterone, testosterone, and estradiol levels at 72 h and 96 h compared to the control group. We expected to find that DNI decreased hormone production because it also inhibited follicle growth in the same treatment group. While DNI dysregulated the expression of steroidogenic regulators at 48 and 96 h, the magnitude of these changes is not enough to explain the decreased hormone levels. From this, we can conclude that DNI is not interfering with the transcription of the steroidogenic regulators. Instead, DNI may interfere with the translation or enzymatic activity of the steroidogenic regulators. Interestingly, the lower concentrations of DNI did not affect steroid hormone secretion despite causing follicle rupture at a comparable rate to the follicles treated with the highest concentration of DNI. Even after rupturing, the theca and granulosa cells remained healthy enough to secrete the steroid hormones. Based on these findings, we think the follicles are not rupturing in response to changes in hormone levels. Rather, the mechanism of follicle rupture is unrelated to follicle growth and steroid hormone synthesis.

Follicle growth is dependent on estrogen signaling, which is mediated by ESR1 and ESR2. We found that IMI increased the expression of *Esr1* and decreased the expression of *Esr2* at 48 h and decreased the expression of *Esr1* at 96 h compared to the control. We do not know the significance of these changes because, to our knowledge, other studies have not reported on the effects of neonicotinoids on estrogen receptor expression.

The survival of an antral follicle depends on the proper balance of pro-apoptotic and anti-apoptotic factors. The *Bax/Bcl2* expression ratio is a reliable indicator of follicular health, and other pesticides have been shown to cause follicle toxicity by altering the expression of these genes [38]. We measured *Bax* (pro-apoptotic) and *Bcl2* (anti-apoptotic) expression to see if they explain the IMI- and DNI-induced changes in follicle growth. IMI did not cause the aberrant expression of *Bax* or *Bcl2*. At all three timepoints, *Bax* and *Bcl2* expression either changed in the same direction or remained unchanged in the IMI-exposed follicles compared to the control. These expression ratios are consistent with the growth data that show that IMI did not interfere with antral follicle growth. DNI induced variable changes to *Bax* and *Bcl2* expression. *Bax* was upregulated at 48 h and downregulated at 72 h in the DNI-exposed follicles compared to the control. At 96 h, *Bax* was downregulated in the lowest treatment group and upregulated in the highest treatment group compared to the control. At 96 h, *Bcl2* was downregulated in the DNI-exposed follicles compared to the control. The significant increase in *Bax* expression at 48 and 96 h is consistent with the growth and hormone data, which show growth inhibition and decreased hormone synthesis in the DNI-exposed follicles compared to the control. Interestingly, we initially expected that *Bax* and *Bcl2* expression would be affected in the follicles treated with the lower concentrations of DNI because many of the follicles ruptured in culture. In fact, our results indicate that the *Bax*/*Bcl2* ratio was not affected, which supports our hypothesis that the follicles in the lower treatment groups maintained sex steroid hormone production because the cells remained viable after the follicle ruptured.

In summary, IMI and its bioactive metabolite DNI induced several different effects on mouse antral follicles in the culture. DNI, but not IMI, inhibited follicle growth and caused follicles to rupture in the culture compared to the control. IMI caused follicles to secrete more progesterone, whereas DNI caused follicles to secrete less testosterone, progesterone, and estradiol at 72 h and 96 h compared to the control. IMI and DNI also differed in how they affected the expression of steroidogenic regulators, estrogen receptors, and apoptotic factors. Our study emphasizes the importance of including intermediate metabolites in toxicology studies, since small changes in chemical structure can result in significant differences in exposure outcomes.

## Figures and Tables

**Figure 1 toxics-11-00349-f001:**
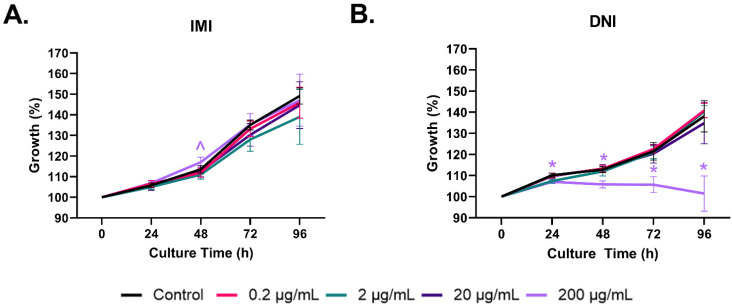
Effects of IMI (**A**) and DNI (**B**) on antral follicle growth. Antral follicles were treated with a vehicle (DMSO or water), IMI, or DNI (0.2–200 µg/mL) for 96 h. Follicle growth was measured every 24 h. The graphs indicate means ± SEMs from 3–10 separate cultures (8–12 follicles per treatment per culture). * *p* ≤ 0.05; ^ 0.05 < *p* ≤ 0.10.

**Figure 2 toxics-11-00349-f002:**
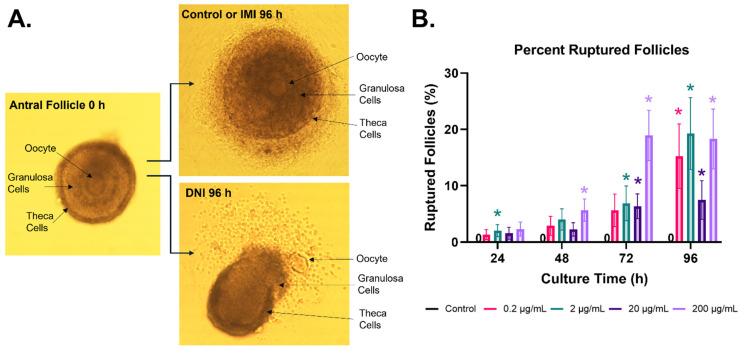
Effects of DNI on follicle morphology. Antral follicles were treated with a vehicle (DMSO) or DNI (0.2–200 µg/mL) for 48, 72, or 96 h. Follicle morphology was monitored every 24 h. The photographs (**A**) are representative images of follicles (20× objective). The proportion of follicles that ruptured in each DNI treatment group was calculated as a percentage (**B**). The zeros within each timepoint indicate that none of the follicles in the control groups ruptured in culture. The graph indicates means ± SEMs from 3–10 separate cultures (8–12 follicles per treatment per culture). * *p* ≤ 0.05.

**Figure 3 toxics-11-00349-f003:**
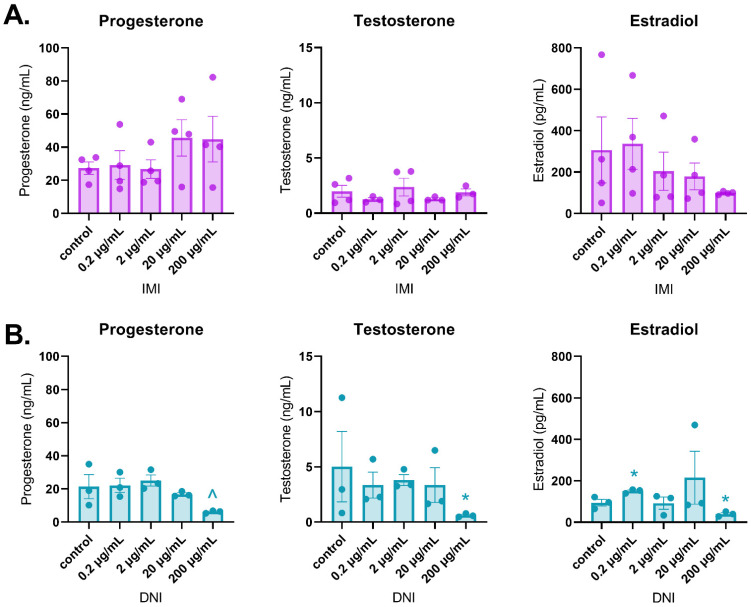
Effects of IMI (**A**) and DNI (**B**) on follicular sex steroid hormone levels after 48 h in culture. Antral follicles were treated with a vehicle (DMSO or water), IMI, or DNI (0.2–200 µg/mL) for 48 h. After the culture period, media from each treatment were pooled and subjected to ELISAs for testosterone, progesterone, and estradiol. The graphs indicate means ± SEMs from 3–5 separate cultures (8–12 follicles per treatment per culture). * *p* ≤ 0.05; ^ 0.05 < *p* ≤ 0.10.

**Figure 4 toxics-11-00349-f004:**
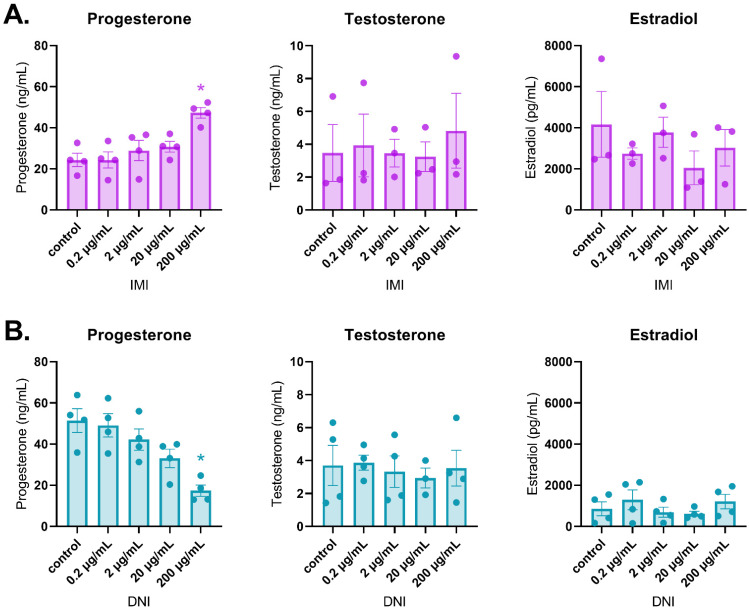
Effects of IMI (**A**) and DNI (**B**) on follicular sex steroid hormone levels after 72 h in culture. Antral follicles were treated with a vehicle (DMSO or water), IMI, or DNI (0.2–200 µg/mL) for 72 h. After the culture period, media from each treatment were pooled and subjected to ELISAS for testosterone, progesterone, and estradiol. The graphs indicate means ± SEMs from 3–5 separate cultures (8–12 follicles per treatment per culture). * *p* ≤ 0.05.

**Figure 5 toxics-11-00349-f005:**
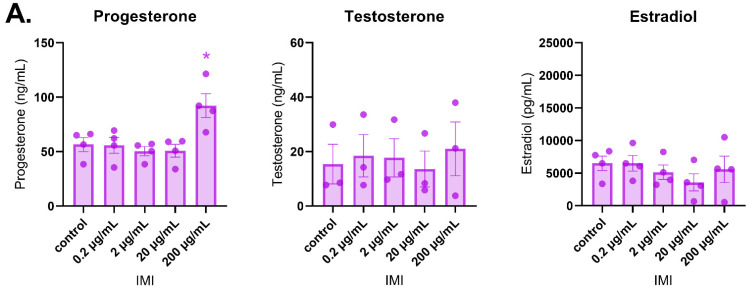
Effects of IMI (**A**) and DNI (**B**) on follicular sex steroid hormone levels after 96 h in culture. Antral follicles were treated with a vehicle (DMSO or water), IMI, or DNI (0.2–200 µg/mL) for 96 h. After the culture period, media from each treatment were pooled and subjected to ELISAs for testosterone, progesterone, and estradiol. The graphs indicate means ± SEMs from 3–5 separate cultures (8–12 follicles per treatment per culture). * *p* ≤ 0.05.

**Figure 6 toxics-11-00349-f006:**
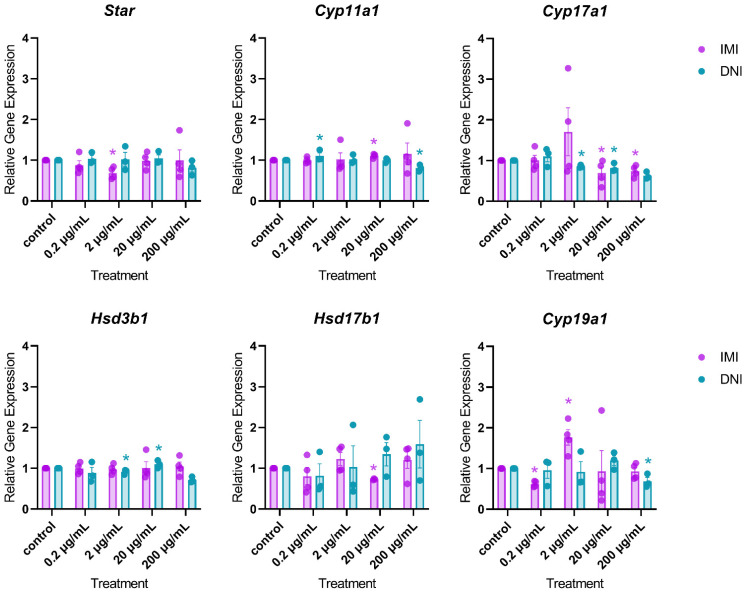
Effects of IMI and DNI on the gene expression of steroidogenic regulators in antral follicles treated with a vehicle (DMSO or water), IMI, or DNI (0.2–200 µg/mL) for 48 h. After the culture period, RNA was extracted from snap-frozen follicles, reverse-transcribed to cDNA, and subjected to qPCR. The graphs indicate means ± SEMs from 3–5 separate cultures (8–12 follicles per treatment per culture). * *p* ≤ 0.05.

**Figure 7 toxics-11-00349-f007:**
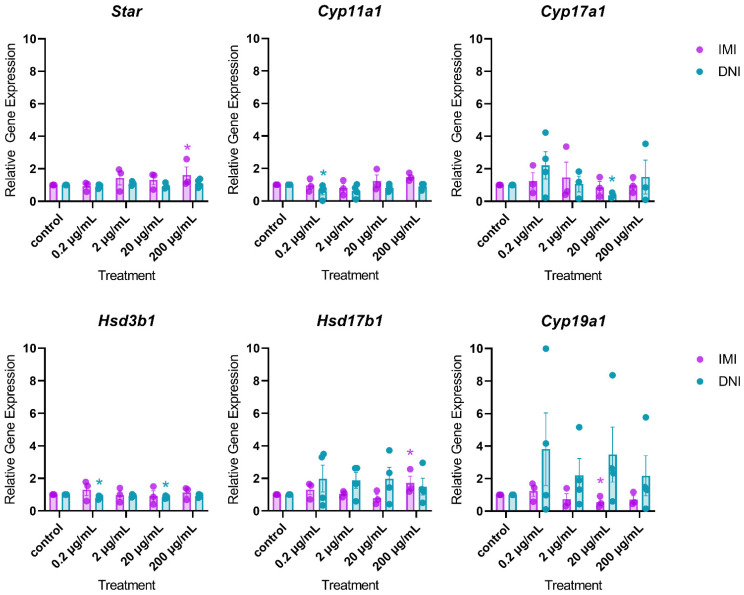
Effects of IMI and DNI on the gene expression of steroidogenic regulators in antral follicles treated with a vehicle (DMSO or water), IMI, or DNI (0.2–200 µg/mL) for 72 h. After the culture period, RNA was extracted from snap-frozen follicles, reverse-transcribed to cDNA, and subjected to qPCR. The graphs indicate means ± SEMs from 3–5 separate cultures (8–12 follicles per treatment per culture). * *p* ≤ 0.05.

**Figure 8 toxics-11-00349-f008:**
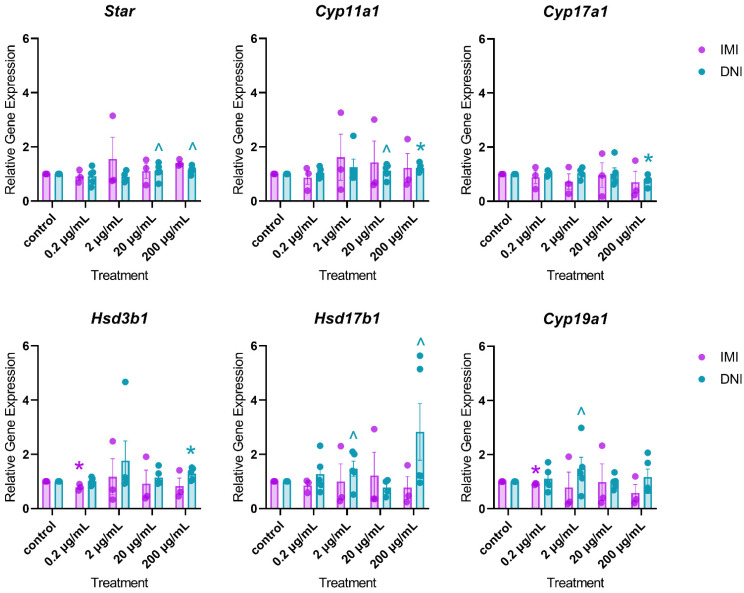
Effects of IMI and DNI on the gene expression of steroidogenic regulators in antral follicles treated with a vehicle (DMSO or water), IMI, or DNI (0.2–200 µg/mL) for 96 h. After the culture period, RNA was extracted from snap-frozen follicles, reverse-transcribed to cDNA, and subjected to qPCR. The graphs indicate means ± SEMs from 3–5 separate cultures (8–12 follicles per treatment per culture). * *p* ≤ 0.05; ^ 0.05 < *p* ≤ 0.10.

**Figure 9 toxics-11-00349-f009:**
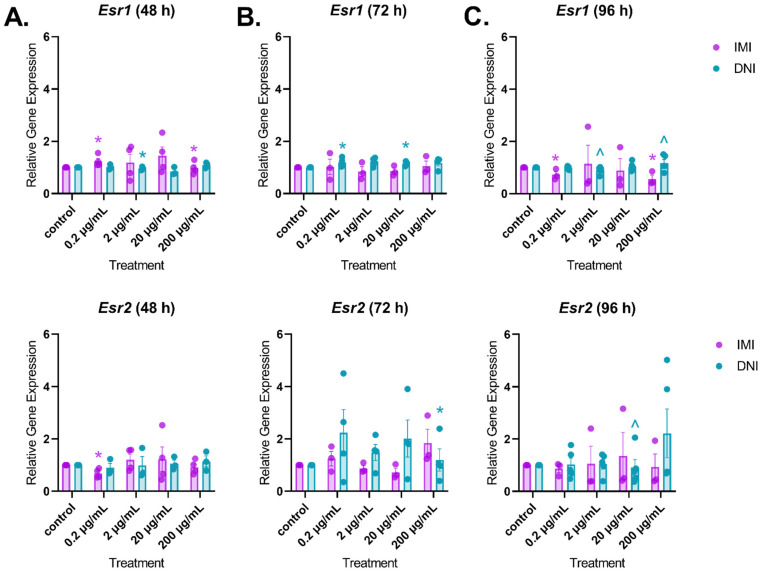
Effects of IMI and DNI on the gene expression of *Esr1* and *Esr2* in antral follicles treated with a vehicle (DMSO or water), IMI, or DNI (0.2–200 µg/mL) for 48 (**A**), 72 (**B**), and 96 (**C**) h. After the culture period, RNA was extracted from snap-frozen follicles, reverse-transcribed to cDNA, and subjected to qPCR. The graphs indicate means ± SEMs from 3–5 separate cultures (8–12 follicles per treatment per culture). * *p* ≤ 0.05; ^ 0.05 < *p* ≤ 0.10.

**Figure 10 toxics-11-00349-f010:**
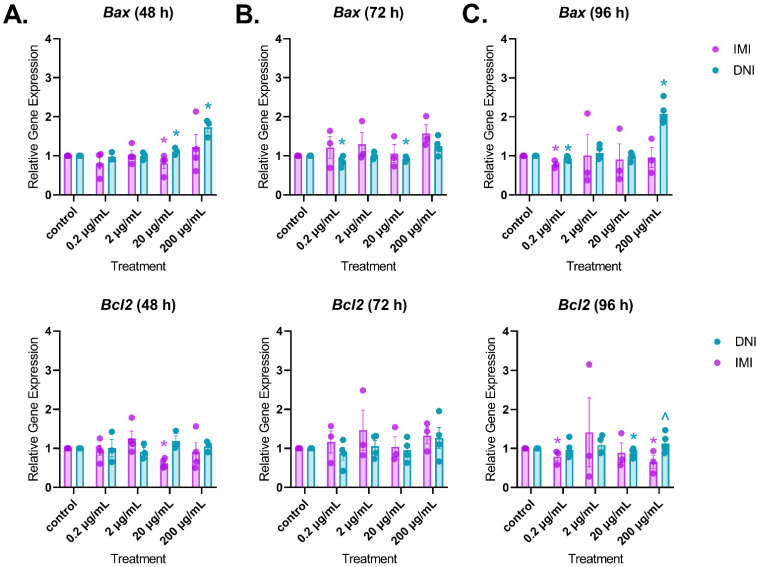
Effects of IMI and DNI on the gene expression of apoptotic factors *Bax* and *Bcl2* in antral follicles treated with a vehicle (DMSO or water), IMI, or DNI (0.2–200 µg/mL) for 48 (**A**), 72 (**B**), and 96 (**C**) h. After the culture period, RNA was extracted from snap-frozen follicles, reverse-transcribed to cDNA, and subjected to qPCR. The graphs indicate means ± SEMs from 3–5 separate cultures (8–12 follicles per treatment per culture). * *p* ≤ 0.05; ^ 0.05 < *p* ≤ 0.10.

**Table 1 toxics-11-00349-t001:** Genes and primer sequences used in qPCR.

Gene Name	Symbol	Forward Sequence	Reverse Sequence
Beta actin	*βactin*	GGGCACAGTGTGGGTGAC	CTGGCACCACACCTTCTAC
Steroidogenic acute regulatory protein	*Star*	CAGGGAGAGGTGGCTATGCA	CCGTGTCTTTTCCAATCCTCTG
Cytochrome P450 cholesterol side-chain cleavage	*Cyp11a1*	AGATCCCTTCCCCTGGTGACAATG	CGCATGAGAAGAGTATCGACGCATC
Cytochrome P450 steroid 17-a-hydroxylase 1	*Cyp17a1*	CCAGGACCCAAGTGTGTTCT	CCTGATACGAAGCACTTCTCG
Cytochrome P450 aromatase	*Cyp19a1*	CATGGTCCCGGAAACTGTGA	GTAGTAGTTGCAGGCACTTC
3b-Hydroxysteroid dehydrogenase 1	*Hsd3b1*	CAGGAGAAAGAACTGCAGGAGGTC	GCACACTTGCTTGAACACAGGC
17b-Hydroxysteroid dehydrogenase 1	*Hsd17b1*	AAGCGGTTCGTGGAGAAGTAG	ACTGTGCCAGCAAGTTTGCG
Estrogen receptor 1 (alpha)	*Esr1*	AATTCTGACAATCGACGCCAG	GTGCTTCAACATTCTCCCTCCTC
Estrogen receptor 2 (beta)	*Esr2*	GGAATCTCTTCCCAGCAGCA	GGGACCACATTTTTGCACTT
B cell lymphoma 2	*Bcl2*	ATGCCTTTGTGGAACTATATGGC	GGTATGCACCCAGAGTGATGC
Bcl2-associated X protein	*Bax*	TGAAGACAGGGGCCTTTTTG	AATTCGCCGGAGACACTCG

## Data Availability

Not applicable.

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
