# Peer review of "Imidacloprid and Its Bioactive Metabolite, Desnitro-Imidacloprid, Differentially Affect Ovarian Antral Follicle Growth, Morphology, and Hormone Synthesis In Vitro"

_toxics, 2023, doi:10.3390/toxics11040349_

Round 1

Reviewer 1 Report

The manuscript “Imidacloprid and its Bioactive Metabolite Desnitro-imidacloprid Differentially Affect Ovarian Antral Follicle Growth, Morphology, and Hormone Synthesis In Vitro.” investigated the ovarian toxicity mechanisms induced by IMI and its metabolite DNI. The results demonstrated that IMI and DNI can differentially affect antral follicle growth and steroid genesis in culture. The manuscript has a message of interest, and it will also draw attention to the toxicity mechanisms of neonicotinoids and their metabolites to mammals. This is an interesting paper thus my recommendation is to accept with major revisions.

1.     What is the basis for the selection of exposure concentrations, the concentration of “200 mg/L” seems to be much higher than environmental exposure concentrations. Please give a reasonable explanation.

2.     Line 89: DNI is one of many intermediate metabolites that are further broken down to the terminal metabolites 6-chloronicotinic acid (6CNA) and 6-hydroxynicotinic acid (6HNA) Tissue disposition experiments conducted in Wister rats indicate that IMI, 6CNA and 6HNA remained in the body for at least 48 h after a single oral exposure. Furthermore, all three of these compounds were detectible in the 70 ovaries of the female animals. Why only select DNI as target metabolites in this study. Please give more specify reason in introduction.

3.     Line 101: section 2.1 Please add the purity of IMI and DNI. Whether the impurities present in the standard will negatively affect the ovarian antral follicle growth and hormone synthesis?

4.     Line 140: add a space between numbers and letters. And change ml to mL. Please check the full text.

5.     Line172: please change “10μl” to “10 μL”, Please check the full text.

6.     Line183-188: In the chapter of gene expression and qpCR verification, 17 key genes were selected in this paper. It is suggested to add explanations in the introduction, why are these 17 genes related to antral follicle growth and hormone synthesis.

7.     Line 200:Result, please add the p, F and df value in result of analysis of variance.

8.     Please add the unit of y-axis in Figure 1 and Figure 2.

9.     Line 226-231: DNI with the highest concentration (200 mg/L) significantly affects sex steroid hormone levels compared to control, and my concern is that this not representative level.

10.  It would be better adding the timepoint (48 h, 72 h, or 96h) in Figure 9 and 10.

11.  Please use the same format of reference, such as 8 and 16 should use the abbreviation of name of journal.

12.  The authors need to take notes in the revision stage and cite more relevant references. I recommend comparison of the results found in this study with the ones found in the relevant literature in discussion section.

Reviewer 2 Report

This manuscript describes an extensive dataset exploring the in vitro potential and mode(s) of action of a widely used insecticide, imidacloprid (IMI) and one of its putative metabolites (desnitro-imidocloprid, DNI) to adversely impact mouse ovarian follicle growth, morphology and hormone synthesis.  An overarching objective was to further inform the potential reproductive toxicity risk potent in IMI.

Although the toxicological-basis of an hypothesis addressing a potential ovarian target of reproductive toxicity potential is reasonably extrapolated from existing IMI and DNI toxicity data, the real-world risk relevance of the hypothesis would substantially benefit from an expanded analysis of its dose-exposure plausibility.  Existing IMI data indicate that DNI is likely a very minor metabolite relative to its overall mammalian metabolism, although it also may be present for potential human exposure as an environmentally-formed IMI metabolite.  Importantly, the manuscript needs to further describe how potential IMI/DNI exposures (doses) likely experienced by humans through consumption of food and/or drinking water contaminated with IMI and/or DNI compare to the in vitro test concentrations used in this study in order to provide dosimetric plausibility to the overall hypothesis.   This reviewer suspects the margin-of-exposure is likely very large in that chronic drinking water exposures to IMI have been estimated as approximately 0.079 ug/kg/day (California Dept. Pesticide Regulation, April 13, 2021), and the resulting ovarian IMI/DNI concentrations from such a dose are likely many orders of magnitude below even the lowest tested concentration of 0.2 ug/ml IMI or DNI (assumed 0.2 ug/g tissue equivalent) of the current study.  In addition, it should be noted that the regulatory (EPA) human risk assessment of IMI is based on development neurotoxicity (i.e., it is the overall most sensitive response observed in animal toxicity), and that even though ovarian effects have been noted at high IMI in vivo doses, IMI did not adversely impact reproductive function in rats.  The overall risk assessment indicates that IMI reproductive toxicity potential is low and would only occur at doses above those eliciting other toxicity(s) of concern.  These dose-exposure (margin-of-exposure) data are thus a substantial uncertainty in the value of the experimental data in plausibly informing the IMI human health reproductive risk potential.

Another substantial uncertainty associated with the interpretation of the risk potential the current data is that, with the exception the follicle growth and rupture data, almost all of the other examined data lack evidence of a dose-dependence, consistent evidence of time-dependent progression, and effects for the hormonal endpoints are largely limited to the highest tested concentration of 200 ug/ml (likely vastly above any ovarian concentrations resulting from real-world in vivo environmental IMI/DNI exposures).  Given these variables, it is entirely unclear if the current dataset reasonably informs the human reproductive health risks of IMI/DNI.

Finally, the overall conclusion (l.319-421) that “…our results indicate neonicotinoids are reproductive toxicants and more studies are required to characterize neonicotinoid toxicity in the female reproductive system” is far too speculative and over-reaching to neonicotinoids in general.  The overall data appear at best to only weakly indicate that high ovarian follicle IMI/DNI concentrations far exceeding those plausibly experienced by humans may potentially impact follicle function.

Specific comments:

l.59-61:  Ref 8 appears to suggest that chlorinated water treatment enables formation of chlorinated metabolites of DNI present in untreated water, but is not responsible for formation of DNI itself.

l.66-68:  Existing metabolism data indicates DNI is a very minor in vivo metabolite of IMI.

l.70-71:  It appears DNI was not detected or not examined for in Ref 11?

l.81-82:  What IMI dose(s) elicited the described ovarian effects in the rat 90-day study, and how do they compare to likely realistic human IMI exposures?

l.91-92:  What is reference supporting this statement?; what dose(s) of IMI were used to detect IMI and DNI in ovaries?; and importantly, how did the detected concentrations (if measured) compare to the test concentrations in this study?

l.124:  Antral follicles cannot be described as a “220 to 400 uM” concentration

Fig. 1B:  The severe effect of 200 ug/ml DNI on follicle growth (essentially halts growth) is likely a substantial confounder to any meaningful toxicological risk interpretation.

Figs 2-10:  As per general comment above, almost all of the data presented in these figures lack dose-dependence, lack time-dependent progression, and hormonal findings are largely limited to the highest tested concentration of IMI/DNI.  Consequently, it is challenging to reasonably attribute most of the findings to treatment-related effects.

Author Response

The authors thank the reviewers for their time and insightful input. We carefully read and considered all the valuable comments and provided specific responses to each comment below All changes to the manuscript have been highlighted in yellow in the revised upload.

Comments and Suggestions for Authors:

This manuscript describes an extensive dataset exploring the in vitro potential and mode(s) of action of a widely used insecticide, imidacloprid (IMI) and one of its putative metabolites (desnitro-imidocloprid, DNI) to adversely impact mouse ovarian follicle growth, morphology and hormone synthesis.  An overarching objective was to further inform the potential reproductive toxicity risk potent in IMI.

Although the toxicological-basis of an hypothesis addressing a potential ovarian target of reproductive toxicity potential is reasonably extrapolated from existing IMI and DNI toxicity data, the real-world risk relevance of the hypothesis would substantially benefit from an expanded analysis of its dose-exposure plausibility.  Existing IMI data indicate that DNI is likely a very minor metabolite relative to its overall mammalian metabolism, although it also may be present for potential human exposure as an environmentally-formed IMI metabolite.  Importantly, the manuscript needs to further describe how potential IMI/DNI exposures (doses) likely experienced by humans through consumption of food and/or drinking water contaminated with IMI and/or DNI compare to the in vitro test concentrations used in this study in order to provide dosimetric plausibility to the overall hypothesis.   This reviewer suspects the margin-of-exposure is likely very large in that chronic drinking water exposures to IMI have been estimated as approximately 0.079 ug/kg/day (California Dept. Pesticide Regulation, April 13, 2021), and the resulting ovarian IMI/DNI concentrations from such a dose are likely many orders of magnitude below even the lowest tested concentration of 0.2 ug/ml IMI or DNI (assumed 0.2 ug/g tissue equivalent) of the current study.  In addition, it should be noted that the regulatory (EPA) human risk assessment of IMI is based on development neurotoxicity (i.e., it is the overall most sensitive response observed in animal toxicity), and that even though ovarian effects have been noted at high IMI in vivo doses, IMI did not adversely impact reproductive function in rats.  The overall risk assessment indicates that IMI reproductive toxicity potential is low and would only occur at doses above those eliciting other toxicity(s) of concern.  These dose-exposure (margin-of-exposure) data are thus a substantial uncertainty in the value of the experimental data in plausibly informing the IMI human health reproductive risk potential.

Another substantial uncertainty associated with the interpretation of the risk potential the current data is that, with the exception the follicle growth and rupture data, almost all of the other examined data lack evidence of a dose-dependence, consistent evidence of time-dependent progression, and effects for the hormonal endpoints are largely limited to the highest tested concentration of 200 ug/ml (likely vastly above any ovarian concentrations resulting from real-world in vivo environmental IMI/DNI exposures).  Given these variables, it is entirely unclear if the current dataset reasonably informs the human reproductive health risks of IMI/DNI.

               Finally, the overall conclusion (l.319-421) that “…our results indicate neonicotinoids are reproductive toxicants and more studies are required to characterize neonicotinoid toxicity in the female reproductive system” is far too speculative and over-reaching to neonicotinoids in general.  The overall data appear at best to only weakly indicate that high ovarian follicle IMI/DNI concentrations far exceeding those plausibly experienced by humans may potentially impact follicle function.

Response: We thank the reviewer for their comments. Our research goals are to fill a substantial gap in the literature regarding the effects of neonicotinoid pesticides and their metabolites on ovarian physiology. In the specific comments below, we have explained why we prioritize DNI over other IMI metabolites and we address the reviewers’ concerns regarding dose and time- dependence.

Importantly, we emphasize that this paper is not intended for use in any regulatory or risk assessment capacity.

One of the reasons we are interested in neonicotinoids is because they are used in diverse industries (agriculture, home gardening, and veterinary medicine). As such, human exposure varies drastically based on geographical location and occupation. When designing our study, we referenced published biomonitoring literature as well as the Centers for Disease Control and Prevention’s NHANES database, which is one of the largest and most inclusive exposure databases in the United States. Importantly, this is an in vitro study where the aim is to directly expose ovarian follicles to IMI and DNI and compare their effects. It is not possible to extrapolate exactly how much IMI reaches a single ovarian follicle. Thus, we opted to use a wide range of concentrations that span the spectrum of human exposure. We agree that 200 ug/ml likely exceeds ovarian exposure to neonicotinoids. We included this dose to elicit an exaggerated response to the parent compound and the metabolite so we can differentiate their effects on antral follicle physiology. For example, we show that IMI increases follicular progesterone secretion, whereas DNI decreases follicular progesterone secretion. Although this dose may not be relevant for regulatory and risk assessment purposes, it is relevant and meaningful to know that small structural changes in neonicotinoids can induce very different outcomes on antral follicle physiology. We added more rationale for our selected doses to the manuscript.  

Specific comments:

  1. 59-61:  Ref 8 appears to suggest that chlorinated water treatment enables formation of chlorinated metabolites of DNI present in untreated water but is not responsible for formation of DNI itself.

Response: We thank the reviewer for their comment. We have corrected our statements and references regarding the formation of DNI in the environment.

  1. 66-68:  Existing metabolism data indicates DNI is a very minor in vivo metabolite of IMI.

Response: We thank the reviewer for their comment. We agree that DNI is one of many metabolites in the breakdown of IMI. We chose DNI because of the possibility that it can cause mammalian toxicity, not because of the amount that is formed. The focus of our experiments is to characterize the effects of neonicotinoids on the mammalian ovary. We prioritized DNI because it has been shown to be more toxic to mammals than IMI and other metabolites. DNI has been shown to bind mammalian nicotinic acetylcholine receptors in a stable manor with very high affinity. In the nervous system, DNI can activate these receptors as efficiently as acetylcholine, which is their endogenous ligand. We clarified these points in the revised manuscript.

  1. 70-71:  It appears DNI was not detected or not examined for in Ref 11?

Response: We thank the reviewer for their comment. DNI was not examined in Ref 11. However, the metabolites that were detected in the ovaries (6CNA and 6HNA) are downstream of DNI metabolism.  We clarified the reasons for our focus on DNI in the revised manuscript.

  1. 81-82:  What IMI dose(s) elicited the described ovarian effects in the rat 90-day study, and how do they compare to likely realistic human IMI exposures?

Response: We thank the reviewer for their comment and have added more details about this study in our revision.

  1. 91-92:  What is reference supporting this statement?; what dose(s) of IMI were used to detect IMI and DNI in ovaries?; and importantly, how did the detected concentrations (if measured) compare to the test concentrations in this study?

Response: We thank the reviewer for their comment. We have added more references and supporting details in the revision.

  1. 124:  Antral follicles cannot be described as a “220 to 400 uM” concentration

Response: We thank the reviewer for their comment and have corrected uM to um.

  1. 1B:  The severe effect of 200 ug/ml DNI on follicle growth (essentially halts growth) is likely a substantial confounder to any meaningful toxicological risk interpretation.

Response: We thank the reviewer for their comment. We do not intend for the data in our paper to be used in regulatory practice or risk assessment. The purpose of our paper is to compare the effects of a parent compound (IMI) to its bioactive metabolite (DNI) on ovarian antral follicles. We agree that 200 ug/ml DNI is halting antral follicle growth. We also show that the same concentration of IMI does not halt antral follicle growth. Our interpretation of these findings is that IMI and DNI are, in fact, differentially affecting antral follicles. It is meaningful to know that minor metabolites can differentially affect ovarian physiology compared to the parent compound.

  1. Figs 2-10:  As per general comment above, almost all of the data presented in these figures lack dose-dependence, lack time-dependent progression, and hormonal findings are largely limited to the highest tested concentration of IMI/DNI.  Consequently, it is challenging to reasonably attribute most of the findings to treatment-related effects.

Response: We thank the reviewer for their comment. We agree that some of our data lack linear or monotonic dose dependence. Reproductive and endocrine toxicants often do not exhibit monotonic dose responses (Vandenberg Et al. Hormones and endocrine-disrupting chemicals: low-dose effects and nonmonotonic dose responses. Endocr Rev. 2012 Jun;33(3):378-455. doi: 10.1210/er.2011-1050. Epub 2012 Mar 14. PMID: 22419778; PMCID: PMC3365860.). Furthermore, we use ovarian follicles from CD-1 mice, which are an outbred mouse strain with inherently variable gene expression that more closely represents human genetic variability. We use qPCR to identify changes in expression that are of large enough magnitude to support our growth, morphology, and hormone data. We agree with the reviewer that our data suggest that IMI and DNI are not acting at the transcriptional level. However, it is meaningful to know that IMI and DNI affect antral follicle growth, morphology, and hormone secretion differently. Finally, we want to point out that all concentrations of DNI caused antral follicles to rupture in culture. This is a significant and unique manifestation of antral follicle toxicity in culture that highlights the importance of including minor metabolites in toxicology studies.

Reviewer 3 Report

Reviewer comments Toxics

 Dear corresponding author

Jodi A. Flaws

Toxics

Manuscript ID: toxics-2264769

Title:

“Imidacloprid and its Bioactive Metabolite Desnitro-imidaclo-2 prid Differentially Affect Ovarian Antral Follicle Growth, Mor-3 phology, and Hormone Synthesis In Vitro”

Comments

Abstract

Please briefly re-write the abstract explaining the experiment design and results concisely without prolongation, then end with suggestions and conclusions.

Please delete abbreviations from abstract

L12-16: Please delete these statements from the abstract. “IMI is readily absorbed in the gastrointestinal tract of mammals where it travels to the liver and undergoes phase I biotransformation into a variety of intermediate metabolites such as desnitro-imidacloprid (DNI), a bioactive metabolite shown to be significantly more toxic to mammals than IMI. Studies have shown that IMI reaches the ovaries almost immediately after oral exposure, reduces ovarian weight, and causes ovarian morphological abnormalities.

Not suitable in abstract but for introduction

L23:but not IMI,” delete it

L24-26: “IMI caused follicles to 24 to secrete more progesterone, whereas DNI caused follicles to secrete less testosterone, progesterone, 25 and estradiol compared to control.”

This means that IMI increases while DNI decreases, authors should mention these results instead

L26-27: “IMI and DNI also differed in how they affected the expression of 26 steroidogenic regulators, estrogen receptors, and apoptotic factors.”

Please clarify how differed

Introduction

L44-46: “Insecticides used as seed 44 treatments cannot be washed or peeled off produce because they spread systemically 45 throughout the crop as it matures [4]. Additionally, the chemical leaches out of the seed 46 contaminating agricultural lands and water systems [5].”

Not clear statements. Please clarify, and rearrange them with grammar corrections  

L76-77: “(estradiol, progesterone, and testosterone),” please delete (testosterone)

It is wrong.

Round 2

Reviewer 2 Report

The revised manuscript has not adequately or appropriately responded to reviewer comments. 

Specific comments:

The author response states their objective was to “fill a substantial gap in the literature”.  However, even the lowest tested IMI concentration used in this study is likely associated with an ovary concentration resulting from an extremely high and human non-relevant 20 mg/kg IMI oral dose.  Importantly, as noted in previous review of this manuscript, the State of California has estimated general population drinking water exposures to IMI as approximately 0.08 ug/kg/day; this estimated daily dose is 250,000-fold less than the overall NOAEL (20 mg/kg/day) for IMI-induced alteration of reproduction function, and 125,000-fold lower than the 90-day NOAEL of 10 mg/kg/day ovarian toxicity.  These very large margins of exposure indicate the findings of this study are not filling an important “data gap” in the overall understanding of IMI potential reproductive toxicity. Also important is the observation that a 20 mg/kg oral IMI dose results in clear symptoms of cholinergic neurotoxicity including diarrhea, salivation, dyspnea, piloerection and tremors, and thus functionally-observed neurotoxicity is the driving endpoint of concern for IMI risk assessment, not reproductive toxicity.

The author response also states that the objective of the study was not intended “for use in any regulatory or risk capacity.”  However, this claim is counter to the author conclusion (Discussion, l.461-462) that the data “indicate neonicotinoids can cause reproductive toxicity.”  At best, the data from this study indicate that very high in vitro test concentrations of IMI/DNI are not dosimetrically plausible in ovaries of general population exposed females, and thus provide no support for such a general conclusion. 

The author response that “IMI increases follicular progesterone secretion, whereas DNI decreases follicular progesterone secretion” is not supported by the reported data (Figs 3&4).  In Fig. 3, none of IMI responses were statistically significant and DNI was decreased only at the excessively high and growth inhibiting 200 ug/ml concentration; in Fig. 4., IMI/DNI statistically significant findings were also only seen at the 200 ug/ml (vastly above any realistic human tissue concentration).  Interpretation of such findings is substantially confounded by high-dose induced toxicity.

The author response to reviewer specific point 2 further highlights that IMI/DNI effects of concern for  humans are more reasonably focused on neurotoxicity and not reproductive toxicity.

Author response to point 3 (and revision l.78-86) states that it can “reasonably” be assumed that “ovaries are continuously exposed to IMI and its metabolites” during “mammals” consumption of contaminated food and drinking water.  This conclusion is unwarranted given the very likely many orders of magnitude lower doses associated with “mammals” (humans) exposure to IMI. This of course is further supported by the literature report that the NOAEL for ovarian toxicity was 10 mg/kg and the overall functional reproductive NOAEL in a repeated dose study was 20 mg/kg/day.  See also comment below for: Revision l.154-165.

Author response to point 4 (revision l.118-122).  What is not mentioned in the revision is the critical observation that 10 mg/kg/day was the overall repeated dose (90 day) NOAEL for ovarian toxicity in the cited references.  In addition, the 20 mg/kg/day dose has been reported as lack effects on functional reproductive performance in rats.

Revision l.128-132:  The IMI metabolic pathway (JMPR, 2001) indicates that IMI is not “hydroxylated to DNI”, but rather to 5-hydroxy-IMI which does ultimately flow through a DNI intermediate.  In addition, l.131-132 incorrectly infer from Ref 20 that DNI has been detected in rat ovaries; DNI was not evaluated in the cited study.

Revision l.154-165:  The NHANES database indicates that IMI urine concentrations were less than the limit of detection (0.4 ug/L; 0.0004 ug/ml) in all women 12-59 years of age, further suggesting that the even lowest in vitro test concentration was likely many orders of magnitude greater than tissue concentrations associated with environmental exposures to IMI.  It is highly incorrect to infer that this study used a “wide range of doses that span the spectrum of human exposure”. At best, it might be correct to state that the lowest tested concentration (0.2 ug/ml) might reflect potential ovary concentrations following an extremely high and non-human relevant 20 mg/kg IMI in vivo dose to rats.  The statement that the 200 ug/ml dose “likely exceeds ovarian exposure” is a vast understatement;  if IMI toxicokinetics are linear, the a 200 ug/ml ovarian concentration would originate from an extremely high in vivo dose (if 20 mg/kg is necessary to produce an ovary concentration of 0.2 ug/ml, 20,000 mg/kg would be required to produce a 200 ug/ml concentration).

Author response to point 8:  The Vandenberg etal citation is unconvincing evidence that it is reasonable to assume that IMI/DNI exhibit “low-dose” effects associated with a non-monotonic dose responses reported for endocrine-disrupting chemicals.  First, as noted above, even the lowest tested concentration translates to an in vivo dose that is many orders magnitude higher than any reasonable human exposures (and thus is not a “low” dose), and second, the Vandenberg etal study has been critiqued as containing speculation that is inconsistent with extensive animal toxicology observations and principles (Rhomberg and Goodman, Regul. Toxicol. Pharmacol. 64: 130-133, 2012).  Thus, the lack of evidence for dose-dependence cannot be assumed as evidence of a “low-dose” non-monotonic dose-response, but rather is more reasonably interpreted as a lack of overall treatment related responses (i.e., effect at low dose followed by no effects at intermediate doses with a return to effects at the highest tested dose).

Round 3

Reviewer 2 Report

The revision has been improved with respect to deleting statements in the Discussion-conclusion inferring that the findings infer reproductive  health concerns for IMI/DNI or neonicotinoids in general.  However, because the Introduction notes that apical ovarian toxicity has been reported only in rats, a rationale should be provided as to why the study use mouse follicles.  Second, l.152-156 infers that selected doses "were based on human biomonitoring literature".  It is not clear how this statement was developed.  All of the cited references only report urine concentrations of IMI and/or its metabolite(s).  How were such concentrations translated to the in vitro concentrations of used this study?  Comparison to urine concentrations alone is not appropriate in that urine concentrations do not readily translate to tissue concentrations.  Ref 36 (Harada et al.) at least translates measured urine IMI to a likely human 0.07 ug/kg/day daily dose; such a dose would certainly result in tissue concentrations well below even the lowest tested dose of this study.  Perhaps a clearer (or more defensible) rationale is that the lowest tested concentrations was equal to that reported in rats dosed with 20 mg/kg IMI.

Author Response

The authors thank the reviewers for their time and insightful input. We carefully read and considered all the valuable comments and provided specific responses to each comment below.   All changes to the manuscript have been highlighted in yellow in the revised upload.

The revision has been improved with respect to deleting statements in the Discussion-conclusion inferring that the findings infer reproductive  health concerns for IMI/DNI or neonicotinoids in general.  However, because the Introduction notes that apical ovarian toxicity has been reported only in rats, a rationale should be provided as to why the study use mouse follicles. 

We used mice in the studies because our laboratory uses the mouse as their experimental model.  This is because several structural and functional features of the mouse female reproductive system are conserved across mammalian species.  Many of the same hormones and genes control female reproduction in mice, rats, and humans.  Further, we wanted to use the CD-1 mouse because it is an out-bred strain and it is a model of choice in many reproductive toxicology studies.  By using the mouse, we can compare the effects of IMI on the mouse ovarian follicle with other chemicals that were examined using mouse ovarian follicles.  We added a sentence to the methods to indicate that why we selected the CD-1 mouse as our model.  Specifically, we state:

We elected to use the CD-1 mouse as our model because mice have been used in thousands of studies on the effects of chemicals on female reproduction and we wanted to be able to compare our  results on the effects of IMI and DNI on the mouse follicle with the results from other studies examining different chemicals.  Further, mouse ovarian follicles resemble ovarian follicles from other species because many of the genes and proteins expressed in ovarian follicles are conserved in across mammalian species.

Second, l.152-156 infers that selected doses "were based on human biomonitoring literature".  It is not clear how this statement was developed.  All of the cited references only report urine concentrations of IMI and/or its metabolite(s).  How were such concentrations translated to the in vitro concentrations of used this study?  Comparison to urine concentrations alone is not appropriate in that urine concentrations do not readily translate to tissue concentrations.  Ref 36 (Harada et al.) at least translates measured urine IMI to a likely human 0.07 ug/kg/day daily dose; such a dose would certainly result in tissue concentrations well below even the lowest tested dose of this study.  Perhaps a clearer (or more defensible) rationale is that the lowest tested concentrations was equal to that reported in rats dosed with 20 mg/kg IMI.

We know that urinary concentrations do not necessarily reflect tissue concentrations.  However, most studies suggest that measurements of urinary levels are a good proxy for exposure levels.  Many toxicology studies use human urinary measures of chemicals to estimate human exposure levels.  Although the estimates may not exactly determine tissue concentrations of a chemical, they are good proxy estimates and give us some information about the ballpark range of exposure.  Thus, we would like to keep our rationale about our dose selection in the manuscript.  It is the truth and why we selected our dose ranges.   That said, we also added to our rationale that is that the lowest tested concentrations were equal to that reported in rats dosed with 20 mg/kg IMI per the reviewer’s suggestion. 
